TECHNICAL RELEASE

# Automated management of AWS instances for training

Jorge Buenabad-Chavez[1,*], Evelyn Greeves[1], James P. J. Chong[1] and Emma Rand[1]

1   Department of Biology, University of York, York, YO10 5DD, UK

## ABSTRACT

Amazon Web Services (AWS) instances provide a convenient way to run training on complex 'omics data analysis workflows without requiring participants to install software packages or store large data volumes locally. However, efficiently managing dozens of instances is challenging for training providers.

We present a set of Bash scripts that make it quick and easy to manage Linux AWS instances pre-configured with all the software analysis tools and data needed for a course, and accessible using encrypted login keys and optional domain names. Creating over 30 instances takes 10–15 minutes.

A comprehensive online tutorial describes how to set up and use an AWS account and the scripts, and how to customise AWS instance templates with other software tools and data. We anticipate that others offering similar training may benefit from using the scripts regardless of the analyses being taught.

**Subjects**  Software and Workflows, Bioinformatics, Metagenomics

## STATEMENT OF NEED

In recent years, sequencing technology has advanced to the point where DNA sequencing for 'omics is faster, easier and cheaper than ever before. Consequently, 'omics experiments increasingly produce large (up to terabyte size) datasets and their analysis requires researchers to access both specialist tools and robust high-performance computing (HPC) infrastructures. Metagenomics analyses are particularly resource intensive. This presents a steep learning curve for biologists who may not have any previous experience using HPC or command line tools.

There is therefore a clear demand for training in this area [1, 2]. However, training provision is complicated by the heterogeneity of individuals' computer setups and the many dependencies demanded by software packages. Furthermore, access to HPC clusters varies depending on the institution and field of study.

Cloud computing services, such as Amazon Web Services (AWS), offer a novel way to provide training in genomics and metagenomics analysis workflows without the need for participants to manage complex software installations or store large datasets in their computers. Each participant can be provided with an identical AWS instance (virtual machine) pre-configured with the software and data needed for a course. The Cloud-SPAN project has been using this approach, based on the Data Carpentry [3] model [4], to deliver highly successful genomics and metagenomics courses for almost three years. This model is known as **Infrastructure as a Service** (IaaS) [5], and is rather flexible, allowing **training providers** to configure virtual machines in terms of compute, storage and networking

**Submitted:**   21 February 2024

\*  Corresponding author. E-mail: jorge.buenabad-chavez@york.ac.uk

Preprint submitted at https://doi.org/10.20944/preprints202408.1095.v1

capacities, as well as the data and software analysis tools required by a course. Meanwhile, **cloud providers** are responsible for managing the actual compute, storage and networking hardware resources and virtualisation. IaaS has also been deployed for bioinformatics training on national HPC clusters [6, 7] using OpenStack (Open Source Cloud Computing Infrastructure) [8] for managing hardware resources and virtualisation. **Platform as a Service** (PaaS) is another cloud computing service model that has been successfully used in bioinformatics training [9, 10]. A PaaS comprises a software environment for data management and analysis tasks using a programming language such as Python or R. Examples include Google Colab [11] and Posit [12], a cloud-based RStudio. These environments are readily accessible through a web browser and simplify sharing code and data.

The main advantages of Cloud computing for training based on the Data Carpentry model are low cost and flexibility. There is no need to manage nor invest in hardware resources or physical space. Instead, an instance in the Cloud is first configured with all the data and software tools required by a course. This instance is then configured as a template, or Amazon Machine Image (AMI) in AWS terminology. Then, for each participant in the course, an instance is created from the AMI. Once the course is over, the instances are deleted in order to stop incurring costs. The AMI is typically preserved to serve as the starting point either (1) to create new instances for a new run of the course, or (2) to create a new AMI with updated data or software, or both, through creating an instance, updating the data or software, and creating a new AMI out of the newly configured instance. In addition, it is rather easy to change the capacity of instances in terms of the number of processors, main memory size, and communication bandwidth to match the processing requirements of the analysis tasks to be taught.

However, managing multiple instances through a graphical user interface, such as the AWS Console, is cumbersome and error-prone. As the number of participants increases, the problem is magnified. The nature of running workshops means participants may drop out, join the course late or not turn up, resulting in further manual management being required.

To address this problem, we developed a set of Bash scripts to automate the management of AWS instances for use in training workshops. The scripts automate the creation and deletion of AWS instances and related resources, namely: encrypted login keys, Internet Protocol (IP) addresses, and domain names. We have also developed an accompanying online tutorial [13] detailing how to open and configure an AWS account, and how to install, configure and run the scripts in a *terminal* on Linux, Windows, and MacOS, or in the AWS CloudShell (browser-based) *terminal*. The tutorial assumes that learners have no prior experience with the AWS concepts and tools covered in the tutorial. However, learners are expected to have some experience with both the Linux/Unix *terminal* and Bash shell programming. Windows users need to install and configure the Git Bash *terminal* and Mac users need to install or update the Bash shell as instructed in the *Precourse Instructions* section of the tutorial [14].

We use the scripts to manage Ubuntu Linux AWS instances configured for training in genomics and metagenomics. However, the scripts are broadly applicable to manage instances configured for any training purpose. The tutorial demonstrates how to customise AMI templates with other software tools and data [15].



## THE SCRIPTS AND HOW TO USE THEM

The scripts are listed below. There are three types of scripts. The primary scripts, "`csinstances_*.sh`", are the topline scripts run by the person in charge of managing instances for workshops. The secondary scripts, "`aws_*.sh`", are invoked by the scripts "`csinstances_create.sh`" or "`csinstances_delete.sh`" to either create or delete instances and related resources: login keys, IP addresses, and domain names (if managed). The third script type corresponds to scripts that provide utility functions to the primary and secondary scripts. The only script in this category is "`colours_utils_functions.sh`", which provides text colouring functions and utility functions that validate the invocation and results of the primary and secondary scripts.

The secondary scripts can each be run directly in the same way the primary scripts are run (as described shortly), but this is not recommended except for the purpose of improving a script or troubleshooting a failed step in creating instances and related resources. The section *Troubleshooting* [16] of the tutorial describes the conditions under which we have had to run some secondary scripts directly.

```
aws_domainNames_create.sh       aws_instances_terminate.sh       csinstances_create.sh

aws_domainNames_delete.sh       aws_loginKeyPair_create.sh       csinstances_delete.sh

aws_instances_configure.sh      aws_loginKeyPair_delete.sh       csinstances_start.sh

aws_instances_launch.sh         colour_utils_functions.sh        csinstances_stop.sh
```

## Managing instances for workshops

Before running the scripts, an AWS account and a terminal environment must be configured (as described in the tutorial [17]). Then, prior to creating instances for a workshop, three files must be created and organised as follows:

- **instancesNamesFile.txt** — contains the names of the instances to be created and managed. **Only the name of this file can be changed** if preferred. This file must contain only one instance name per line, and each instance name must start with an alphabetic character followed by alpha-numeric characters, hyphens (-) or underscores (_) only.
- **resourcesIDs.txt** — contains a set of space-separated "key value" pairs that specify the AWS resources to use in creating each instance and related resources. This is the contents of the **resourcesIDs.txt** file we use for the Cloud-SPAN Genomics course [18]:

```
KEYWORD          VALUE examples (Cloud-SPAN's for Genomics course using instance domain names)

                                    ## NB: "key value" pairs can be specified in any order

imageId          ami-07172f26233528178    ## NOT optional: instance template (AMI) id

instanceType     t3.small                 ## NOT optional: processor count, memory size, bandwidth

securityGroupId  sg-0771b67fde13b3899     ## NOT optional: should allow ssh (port 22) communication

subnetId         subnet-00ff8cd3b7407dc83 ## optional: search vpc in AWS console then click subnets

hostZone         cloud-span.aws.york.ac.uk ## optional: specify to use instance domain names

hostZoneId       Z012538133YPRCJ0WP3UZ    ## optional: specify to use instance domain names
```

As shown in this example, a **resourcesIDs.txt** file can have comments in addition to the "key value" pairs to specify. The "key value" pairs can be specified in any order, but each key word must be the first item in a line and its corresponding value the second item in the same line. The key words in the example must be used, but they are NON-case sensitive. The three not optional "key value" pairs must be specified.



All the values are validated. The value of imageId is validated to correspond to an AMI in the user's AWS account or to a public AMI available in the AWS region on which the scripts are running. The value of instanceType is validated to be a valid AWS instance type. The values of securityGroupId, subnetId, `hostZone` and `hostZoneId` are validated to exist in the user's AWS account.

The key word subnetId and its value are optional. If not specified, the scripts will try to obtain a subnetID from the user's AWS account. We have successfully tested the scripts to obtain and use a subnetID running the scripts with a personal AWS account and with an institutional AWS account (see details in the section *Validating the Workshop Environment* below).

The keywords `hostZone` and `hostZoneId` and their values are optional. If specified and valid, each instance will be accessed using a domain name such as the following: **instance01.cloud-span.aws.york.ac.uk**, where **instance01** is an example of a specified instance name and **cloud-span.aws.york.ac.uk** is the base domain name (`hostZone`) in this example. If `hostZone` and `hostZoneId` and their values are not specified, each instance will be accessed using the public IP address or the generic domain name allocated by AWS, which will look like the following: **34.245.22.106** or **ec2-34-245-22-106.eu-west-1.compute.amazonaws.com**.

- **tags.txt** — contains a set of space-separated "key value" pairs to tag instances and related resources upon creation. **This file is optional**. If specified, it must contain **only one** "key value" pair per line. Up to 10 "key value" pairs are processed. Examples:

```
group       BIOL
project     cloud-span
status      prod
pushed_by   manual
```

The three files must be placed inside a directory called **inputs**, and the inputs directory must be placed within at least one other directory whose name you can choose, and to which we refer to as **Workshop Environment (WE)**. We use this directory structure to manage instances for our workshops:

```
courses                         ### you can omit this directory or use other name
  genomics01                    ### workshop/course WE name; you can use other name
    inputs                      ### you CANNOT use other name
      instancesNames.txt        ### you can use other name
      resourcesIDs.txt          ### you CANNOT use other name
      tags.txt                  ### OPTIONAL - you CANNOT use other name
    outputs                     ### created automatically by the scripts - don't modify
  genomics02                    ### another WE: inputs and outputs directories inside
  metagenomics01                ### another WE: inputs and outputs directories inside
  ...
```

The `outputs` directory inside a WE is created automatically by the scripts to store the results of invoking AWS services as described below.

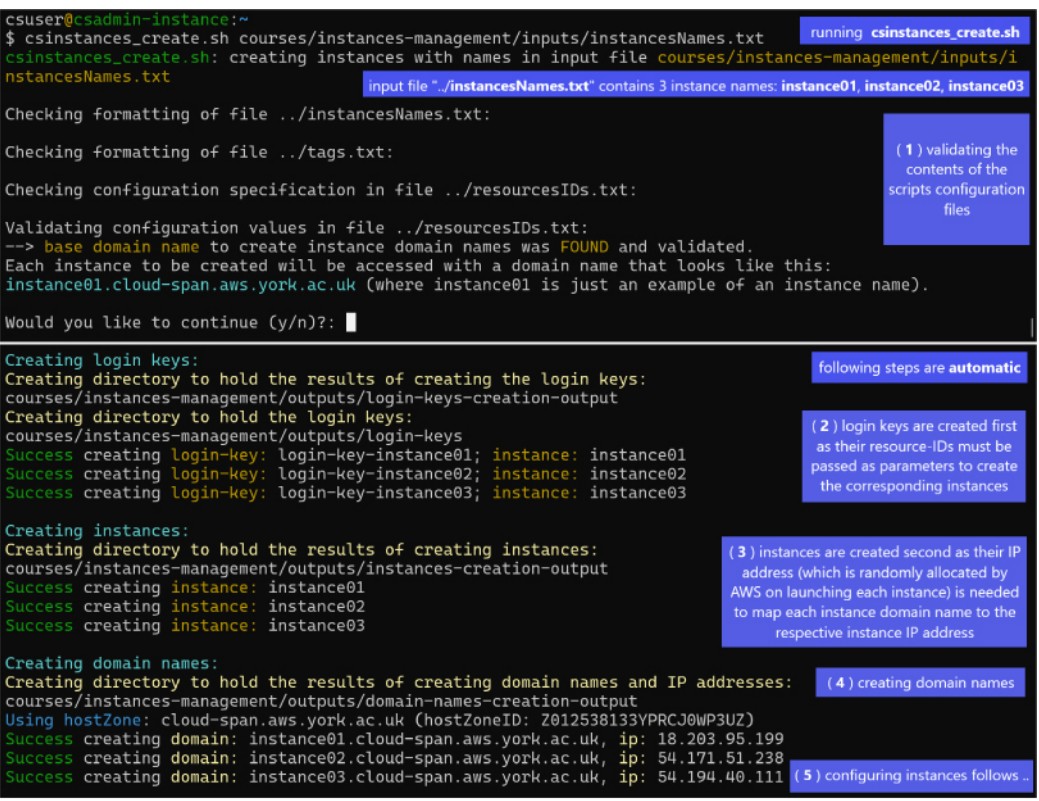

**Figure 1.** Running the script `csinstances_create.sh` (second line) to create three instances with the names specified in the file `courses/.../instancesNames.txt`: **instance01**, **instance02**, and **instance03** (one name per line). Creating instances involves five or four steps depending on whether domain names to access instances are to be managed or not. The run in the figure corresponds to domain names being managed, but only the first four steps are shown. The last step (5) involves configuring each instance both **to enable** the *csuser* account (used by workshop participants) to be logged in and **to change** the instance host name to the instance name (the default host name is the instance IP address). If domain names are not to be managed, the step creating domain names (four in the figure) is not run.

## Running the scripts

Running the scripts requires only the path of the file that contains the names of the instances to create, stop, start or delete. Login keys, IP addresses and domain names used by the instances are created or deleted automatically.

Figure 1 shows a Linux terminal where the script `csinstance_create.sh` has been run thus (second line in the Figure 1):

```
$ csinstances_create.sh      courses/instances-management/inputs/instancesNames.txt
```

Creating instances involves five or four steps depending on whether domain names to access instances are to be managed or not:

- **(1)** validating the contents of the scripts configuration files (***instancesNames.txt*** passed as parameter, **tags.txt** if found, and **resourcesIDs.txt**) as described above and shown in Figure 1. If no problem is found in these files, the option to continue with the configuration detected, regarding managing or not managing domain names to access instances, is displayed for the user to confirm or cancel the run. If there is a problem

with the files, messages (not shown in Figure 1) are displayed to specify the specific problem/s in each file and the run is aborted.

In Figure 1, the configuration detected corresponds to managing domain names to access instances; that is, `hostZone` and `hostZoneId` and valid values were specified and found in the **resourcesIDs.txt** file and hence validated. If `hostZone` and `hostZoneId` are not specified, the option to continue looks like this:

If `hostZone` and `hostZoneId` are not specified, the option to continue looks like this:

```
--> NO base domain name was FOUND.

Each instance to be created will be accessed with the IP address or the generic domain name provided
by AWS, which look like this: 34.245.22.106 or ec2-34-245-22-106.eu-west-1.compute.amazonaws.com.

Would you like to continue (y/n)?:
```

- **(2)** creating login keys.
- **(3)** creating instances, each configured to use one of the login keys created in the previous step.
- **(4)** creating instance domain names as mapped to the respective instance IP addresses (AWS randomly allocates IP addresses to instances when instances are launched for the first time or started after having been stopped) — this step is only run if `hostZone`, `hostZoneId` and valid values are specified in the **resourcesIDs.txt** file.
- **(5)** configuring each instance both to enable the *csuser* account (used by workshop participants) to be logged in and to change the instance host name to the **instance name** (the default host name is the instance IP address) regardless of whether domain names are to be managed or not. This step is not shown in the Figure 1.

To stop or start the instances created with the command above, or to delete them along with all the related resources, you would run the scripts `csinstances_stop.sh`, `csinstances_start.sh` or `csinstances_delete.sh` using the same input file:

```
$ csinstance_stop.sh    courses/instances-management/inputs/instancesNames.txt

$ csinstance_start.sh   courses/instances-management/inputs/instancesNames.txt

$ csinstance_delete.sh  courses/instances-management/inputs/instancesNames.txt
```

## Using instances and customising AMIs

Each instance has two user accounts:  *csuser* and  *ubuntu*. You login to these accounts using the **ssh** program as in the examples below, which correspond to how you would login to those accounts on the instances created in the example in Figure 1, wherein the instance names specified were **instance01**, **instance02**, and **instance03**.

```
ssh -i login-key-instance01.pem csuser@instance01.cloud-span.aws.york.ac.uk

ssh -i login-key-instance01.pem ubuntu@instance01.cloud-span.aws.york.ac.uk

...

ssh -i login-key-instance03.pem csuser@instance03.cloud-span.aws.york.ac.uk

ssh -i login-key-instance03.pem ubuntu@instance03.cloud-span.aws.york.ac.uk
```

Note that instance names are used by the scripts to "label" the names of the corresponding login key files, instance domain names, and other files. Each instance domain name is the previously configured *base domain name* (in our case, `cloud-span.aws.york.ac.uk`) prefixed with the corresponding instance name.

Workshop participants use the **csuser** account — all 'omics data and most software analysis tools are installed in this account.

The **ubuntu** account has superuser privileges. We use it to update system software and software analysis tools that need to be installed at system level. We use the **csuser** account to update 'omics data and software analysis tools that can be installed at user account level. We update an instance as just outlined in order to create a new AMI (virtual machine template) from which to create new updated instances.

### Login to instances when domain names are NOT managed

Assuming that domain names were not managed in the example in Figure 1 (and that instance names were those in the example, i.e., **instance01**, **instance02**, and **instance03**), you would login to the **csuser** and **ubuntu** accounts as follows:

```
ssh -i login-key-instance01.pem csuser@34.245.22.106

ssh -i login-key-instance01.pem ubuntu@34.245.22.106

...
```

The IP address 34.245.22.106 is just an example IP address. The IP address of each instance will vary.

## Customising the login account of workshop participants

The **csuser** account is only available on instances created from a Cloud-SPAN AMI — "cs" in **csuser** stands for Cloud-SPAN project [19, 20]; the Data Carpentry AMI uses the **dcuser** account for workshop participants.

The **ubuntu** account is available on instances created from any AWS Linux Ubuntu AMI, and is the only account that is enabled to be logged in when an instance runs for the first time. This enabling is performed by the AWS service that launches instances. When an instance boots for the first time, that service adds the public key part of the login key created to access the instance to the file `/home/ubuntu/.ssh/authorized_keys`. Thereafter, the private key part of the login key can be used with ssh, as shown above [21], to access the **ubuntu** account. The last step in creating instances (i.e., configuring instances) enables the **csuser** account in each instance to be logged in by copying that file to the **csuser** account in `/home/csuser/.ssh/authorized_keys`. The copy is made by running a Bash script in the **ubuntu** account that is remotely invoked by the script `aws_instances_configure.sh`.

If instead of the **csuser** account, you would like to use an account with a name related to your project, institution, or another element, you can follow the steps below to create a new AMI from which you will create instances with your new user account (see details in the section *Configure an Instance to Become AMI* [15] in the tutorial):

- create an instance using the Cloud-SPAN Genomics AMI and log in as the **ubuntu** user.
- create and initialise your new user account (as described in the tutorial)
- edit the script `/home/ubuntu/bin/usersAuthorizedKeys-activate.sh` to replace the string **csuser** with the name of your new account — this script copies the file `/home/ubuntu/.ssh/authorized_keys` to the **csuser** account.



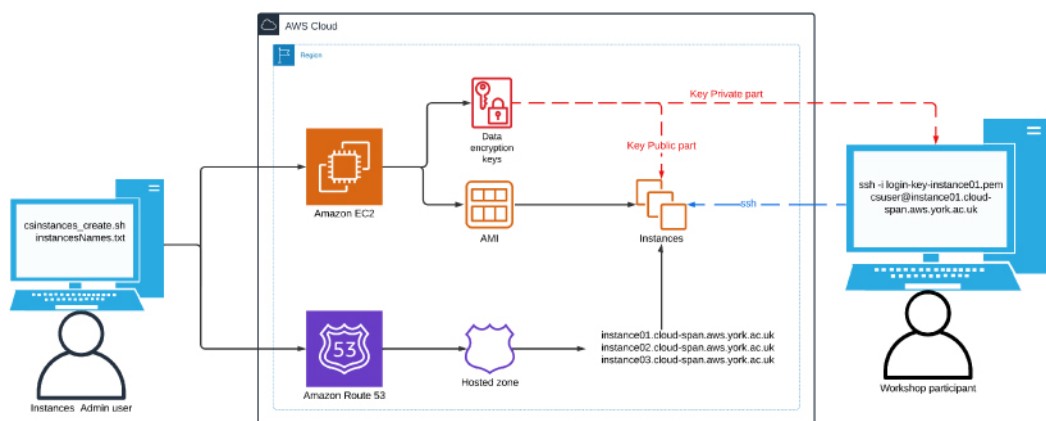

**Figure 2.** The scripts environment: the admin user of the scripts (on the left), the AWS service infrastructure used when domain names to access instances are managed, and a workshop participant accessing **instance01** using *ssh* with the respective private login key part. The Route 53 AWS service is not used when domain names are not managed.

- delete or edit the file /etc/motd (message of the day) — it contains a screen welcome message that includes the Cloud-SPAN name.
- create a new AMI from the updated instance.

The Cloud-SPAN Genomics AMI is 30 GB in size. It has the scripts installed along with a few other scripts that will help you customise your AMI. Two scripts automatically update system software and the Genomics software applications installed. The tutorial describes how to modify these scripts to install other applications and remove the ones installed, as well as how to use another script to automatically increase the size of secondary storage and the file system by up to 2 TB.

Figure 2 shows a scheme of the scripts environment. We have described the left and right elements in that figure: the use of the scripts and the use of instances created with the scripts. How the scripts manage the relevant AWS services is described below.

## THE SCRIPTS DESIGN AND IMPLEMENTATION

AWS services can be managed using (1) the AWS Console, (2) the AWS CLI (command line interface) program, (3) Software Development Kits (SDKs) or libraries with a programming language such as Python, Java, etc., or (4) infrastructure as code (IaC) blueprints. The level of automation increases from the AWS Console to IaC blueprints [22].

The AWS Console is mostly used to open an AWS account, do one-off configurations, and browse the overall state of resources used by an account. The AWS CLI enables you to manage any AWS service but is complex to use on its own; however, combined with shell scripts, it is probably the fastest way to manage AWS services, as shell scripts have been used for decades for managing resources. SDKs are mostly used to develop end-user applications comprising a front-end (mobile or browser-based) user interface and back-end cloud-based server/s and databases. IaC blueprints are used to manage infrastructures (service architectures) that are complex [23], that change often because of continuous improvements, or that require "zero-downtime deployments" where "changes must be made with live traffic" [23]. Basically, a blueprint of the required infrastructure is written as

code in a declarative language specifying "what we want", as opposed to "how to do what we want" (which is typical of procedural languages). On "running" the blueprint using a software tool such as AWS CloudFormation or Terraform, the services making up the infrastructure are created, configured to some extent, and launched. If the blueprint is updated later (for instance, to delete or add more services), the infrastructure will be updated accordingly when the blueprint is run again. As blueprints are simple text files, it is possible to use version control to roll back to a previous version of the infrastructure. It may be convenient and practical to develop a system using the AWS CLI, SDKs and IaC.

The scripts use the AWS CLI to manage instances and related resources. Using Bash scripts and the AWS CLI was a design decision based on convenience and feasibility, given the time constraints we had when confronted with the task of managing multiple AWS instances. We had limited experience with Python AWS SDK (compared to Bash and the AWS CLI) and had not used IAC before. In hindsight, we believe our decision was practical and adequate. The AWS infrastructure we manage (login keys, instances, IP addresses and domain names) is relatively simple and has not changed since we developed the scripts almost three years ago. Also, one of the main goals of the Cloud-SPAN project was to create a solution for managing AWS instances that could be easily shared and taught to others. By using Bash scripting with the AWS CLI, we made the online tutorial simpler and more readily accessible to more people as Bash is more widely known and used than SDKs, Terraform, or other similar tools.

The scripts organisation is straightforward, with most communication between the scripts taking place through shared files. Some scripts perform a fair amount of pattern matching in preparing AWS service requests, processing results requests, and performing other tasks.

### The scripts execution flow — overview

We saw above how to run the scripts `csinstances_create.sh` and `csinstances_delete.sh`: passing the path of the file that contains the names of the instances to be created or deleted (along with related resources). Those two scripts do not invoke the AWS CLI directly; they only invoke the scripts "`aws_*.sh`", passing each such script the file path received, as shown in the main code of `csinstances_create.sh` below. In the code, the file path is in the script variable $1, as `csinstances_create.sh` receives the file path as the first (and only) parameter:

```
check_theScripts_csconfiguration        "$1" || { message "$error_msg"; exit 1; }

aws_loginKeyPair_create.sh              "$1" || { message "$error_msg"; exit 1; }

aws_instances_launch.sh                 "$1" || { message "$error_msg"; exit 1; }

if [ -f "${1%/*}/.csconfig_DOMAIN_NAMES.txt" ]; then ### %/* gets the inputs directory path

     aws_domainNames_create.sh          "$1" || { message "$error_msg"; exit 1; }
fi

aws_instances_configure.sh              "$1" || { message "$error_msg"; exit 1; }

exit 0
```

In the code above, the function `check_theScripts_csconfiguration` (which is in the script file `colour_utils_functions.sh`) is first invoked to validate the contents of the configuration files *instancesNamesFile.txt*, **resourcesIDs.txt**, and **tags.txt** if specified. The scripts

"`aws_*.sh`" are only invoked if that function or the previous script runs successfully; otherwise, `csinstances_create.sh` prints an error message and aborts execution (exit 1;) — unsuccessful runs are discussed below in the section *Validating the Workshop Environment*. The script `aws_domainNames_create.sh` is only invoked if the file `../inputs/.csconfig_DOMAIN_NAMES.txt` exists, which is created if, in validating the **resourcesIDs.txt** file, a base domain name and its host zone ID are found and are valid AWS resources within the AWS account being used.

The code of `csinstances_delete.sh` is similar to the code above but invokes the scripts that delete AWS resources.

### Creating and deleting instances and related resources

Each "`aws_*.sh`" script (except `aws_instances_configure.sh`) can create or delete ***one*** or ***multiple*** AWS resources of a single type. For example, they can delete one or more login keys or one or more instances. Each script invokes the AWS CLI with a specific AWS service request for each instance name specified in the input file received as a parameter. Each script makes a list with the instance names in the input file and then loops through the list. At each iteration, each script invokes the relevant AWS service "for the instance name" in the *current* loop, as outlined next. These are some of the AWS services invoked by the "`aws_*.sh`" scripts:

```
aws ec2 create-key-pair --key-name $loginkey --key-type rsa ...   ### invoked by aws_loginKeyPair_create.sh
aws ec2 run-instances --image-id $resource_image_id ...           ### invoked by aws_instances_launch.sh
aws ec2 delete-key-pair --key-name $loginkey ...                  ### invoked by aws_loginKeyPair_delete.sh
aws ec2 terminate-instances --instance-ids $instanceID ...        ### invoked by aws_instances_terminate.sh
```

The AWS CLI program is called **aws** once installed. The first parameter, ec2 (elastic compute cloud), is the AWS service being invoked; the second parameter, create-key-pair, or run-instances, etc., is the operation being requested; the following parameters are key-value pairs required by the operation. Some values of the key-value pairs are specified as script variables, for example: `$login_key` and `$resource_image_id`. Some of these variables are updated within each loop just before invoking the AWS CLI so that each invocation uses the values (in those variables) that correspond to the relevant instance name.

### Scripts communication

Each invocation of the AWS CLI returns a result that states whether the AWS service request was successful or not along with other information. In the case of a successful request that ***creates*** or ***allocates*** a resource, the result includes the ***resource-id*** assigned by AWS to uniquely identify the resource. As resource-ids are needed to further manage the corresponding resources, for example, to delete them, the scripts store the results of each AWS invocation into a file inside the outputs directory in the WE being used. The name of each file has, as a sub-string, the instance name used to invoke the AWS service to create the resource. This file naming convention enables the other scripts to later recover the resource-id of the resources to delete, stop, or start.

The names of login key files and domain names are managed similarly by including the relevant instance name as a sub-string, as shown in the login examples with ssh shown earlier.



## Configuring, stopping and starting instances

The scripts `aws_instances_configure.sh`, `csinstances_stop.sh`, and `csinstances_start.sh` are organised and work similarly to the other "`aws_*.sh`" scripts that create or delete AWS resources. They build a list of instance names and loop through the list in order to invoke the AWS services required to configure, stop, or start each instance in the list.

However, these scripts are somewhat more complex than the others, as configuring, stopping and starting instances require different handling, depending on whether domain names are managed or not, and these scripts handle both scenarios.

A crucial point for admin users of the scripts is that not managing domain names requires knowing where to find the IP addresses allocated to instances in order to provide them to workshop participants. Each of these IP addresses is saved into a file in the directory **outputs/instances-creation-output** in the WE being used. For example, the contents of this directory after creating three instances (instance01, instance02, instance03) **not** managing domain names would be as shown in this file listing:

```
csuser@csadmin-instance:~
$ ls courses/instances-management/outputs/instances-creation-output/
instance01-ip-address.txt    instance02-ip-address.txt    instance03-ip-address.txt
instance01.txt               instance02.txt               instance03.txt
```

The contents of the files "`instance*-ip-address.txt`" are just the IP addresses of the instances:

```
csuser@csadmin-instance:~
$ cat courses/instances-management/outputs/instances-creation-output/instance*-ip-address.txt
3.252.164.131
34.242.5.243
3.249.248.80
```

The files "`instance??.txt`" are created by the script `aws_instances_launch.sh`; the files "`instance*-ip-address.txt`" are created by the script `aws_instances_configure.sh`. This is because an instance public IP address is not available in the results from invoking `aws_instances_launch.sh` (that is, in the files "`instance??.txt`"), but is available until each instance is actually running, a condition that the script `aws_instances_configure.sh` waits for and detects in order to recover each instance IP address to start configuring the instance, and finally save the IP address into the file *instanceName*-ip-address.txt.

If instances are stopped and eventually started through running `csinstances_stop.sh` and `csinstances_start.sh`, the IP address of each instance will change because AWS randomly allocates IP addresses when instances are launched for the first time or when they are started after having been stopped. Therefore, `csinstances_start.sh` overwrites the contents of the files "*instanceNames*\*-`ip-address.txt`" with the newly allocated IP addresses.

Additionally, the admin user of the scripts does not need to use IP addresses to log in to the respective instances - only workshop participants need to use IP addresses if domain names are not managed. The admin user can instead use the script `lginstance.sh` providing the path of the instance login key file and the user account to login (***csuser*** or ***ubuntu***):

```
csuser@csadmin-instance:~

$ lginstance.sh courses/instances-management/outputs/login-keys/login-key-instance01.pem csuser

  logging you thus:            ### this and the next line are displayed by lginstance.sh

  ssh -i courses/instances-management/outputs/login-keys/login-key-instance01.pem csuser@3.253.59.74

  ...                          ### instance welcome message

csuser@instance01:~           ### instance prompt

$
```

To note, that long command (“`lginstance.sh` …”) is rather easy to enter using the Tab key for the shell to complete the name of the script `lginstance.sh`, the names of the intermediate directories and the name of the login key .pem file. Also, as mentioned above, the host name of an instance is the instance name and not its IP address, regardless of whether domain names are managed or not. These two points will help troubleshoot if domain names are not managed.

### Validating the target workshop environment

We discussed above most of the validation of the contents of the scripts' configuration files: ***instancesNames.txt***, **tags.txt** (if specified), and **resourcesIDs.txt**. The following points will help understand better how the scripts work.

The scripts' configuration files are all validated only the first time any of the scripts is run against a WE; thereafter, only the file ***instancesNames.txt*** is validated (below is outlined why and how you may want to handle multiple ***instancesNames.txt*** files in the same WE). Specifically, once it is determined, validated, and confirmed (by the user of the scripts) whether domain names are to be managed or not, either the file `inputs/.csconfig_DOMAIN_NAMES.txt` or the file `inputs/.csconfig_NO_DOMAIN_NAMES.txt` is created in the WE being used. These files are empty. It is only their existence that is used to drive the execution of the scripts accordingly.

In the file **resourcesIDs.txt**, the key word subnetId and its value are optional and, if they are not specified, the scripts will try to obtain a subnetID from the AWS account being used. As mentioned above, we successfully tested the scripts to obtain and use a **subnetID** running the scripts both with a **personal** AWS account and with an **institutional** AWS account. Our personal account uses the default subnets allocated by AWS. Our institutional account is configured not to use the default subnets but other subnets configured by the Information Technology (IT) Department at our institution. If the scripts cannot obtain a subnetID, they will display a message asking you to search for **vpc** (virtual private cloud) in the AWS Console (in the search box at the top), then click on subnets on the left menu. Copy one of the subnetIDs displayed and paste it into the **resourcesIDs.txt** file. If no subnetID is displayed in the AWS Console, you need to contact your IT department.

In addition to the validation already discussed, the scripts follow an **all-or-nothing** policy in managing resources, as follows. When creating, deleting, stopping, starting, or configuring instances or related resources, any such operation must be possible for *all* the instances (or resources implicitly) specified in the file “*instanceNamesFile.txt*” passed to the scripts. If there is a problem, the scripts cancel their operation. The policy is implemented by checking whether any of the specified instances/resources exists already, that is, whether any of the files containing the AWS resource-id of the specified instances/resources exists in the outputs directory of the target workshop environment. No such file should exist when

creating instances/resources. Conversely, all relevant files should exist when deleting instances/resources, or stopping, starting or configuring instances. The policy was implemented to avoid overwriting the files with AWS resource-ids accidentally, as when this happened the corresponding AWS resources had to be deleted manually using the AWS Console.

To note, you can manage multiple *"instanceNamesFiles.txt"* in the inputs directory of any workshop environment. We do so to handle late registrations and cancellations to our workshops. The section *Unforeseen Instance Management* [24] in the tutorial describes our approach to naming multiple *"instanceNamesFiles.txt"* so we can easily track all the script runs performed in a workshop environment.

## Overview of the online tutorial

How to configure and use the scripts is described in detail in the tutorial *Automated Management of AWS Instances* [13], which covers these main topics:

- How to open an AWS account and how to configure it both with programmatic access with the AWS CLI and with a base domain name from which to create instances (sub) domain names.
- How to configure a ***terminal*** environment with the scripts and the AWS CLI on Linux, MacOS, and Windows (Git Bash) or in the AWS CloudShell, a browser-based Linux Bash *terminal*.
- How to configure and run the scripts to manage instances for a workshop, manage late registrations and cancellations, and do some troubleshooting.
- How to create, manage and configure AMIs, which serve as templates to create AWS instances.
- The organisation and workings of the scripts.

## CONCLUSIONS

We have presented a set of scripts that make it easy and convenient to manage AWS instances for training delivery. Once an AWS account and a *terminal* environment have been configured, only two or three files need to be configured to create and manage instances for a workshop. However, most of the time, only the file ***instancesNames.txt*** needs to be configured with the names of the instances to manage. The file **tags.txt** is optional; if specified, it has to be configured only once and we copy it for all our workshops. Similarly, the file **resourcesIDs.txt** needs to be configured only once for all the workshops that use instances created with the same AMI template and the same instance type (number of processors, memory size, and bandwidth). The configuration of instances can also be managed with AWS *launch templates* [25, 26]: **json** objects that specify the AWS resources to use in creating instances. We did not explore this option while designing the scripts. We consider our solution based on plain text configuration files much simpler to use for the management of multiple instances that we require. Using launch templates with our scripts would involve handling, for each launch template, a local file with the template resource-id (the equivalent to our resourcesIDs.txt file), but also creating and managing the launch templates in the AWS Console or with the AWS CLI.

Configuring an AWS account and a *terminal* environment for use with the scripts is somewhat involved for the various technologies involved, but it must be done only once

and the online tutorial covers all the details. It should take two to four hours, depending on the user's experience, to cover the tutorial along with configuring the AWS account and the *terminal* environment.

We use the scripts to support the delivery of 'omics training, but the scripts will work equally well for AWS Linux Ubuntu instances configured for other purposes.

## AVAILABILITY OF SOURCE CODE AND REQUIREMENTS

- Project name: Cloud-SPAN (https://cloud-span.york.ac.uk/ — https://github.com/Cloud-SPAN).
- Project home page (the *Bash scripts*): https://github.com/Cloud-SPAN/aws-instances.
- Operating system(s): Linux, Windows, MacOS.
- Programming language: Bash Shell.
- Other requirements: Bash version 5.0 or higher. Windows users must install Git Bash. MacOS users must install or update Bash. The online tutorial provides instructions for Windows and MacOS users to do so [14]. The AWS CLI must be installed and configured. The online tutorial provides instructions for Linux, Windows and MacOS users to install and configure the AWS CLI [27].
- License: MIT Licence.
- RRID:SCR_025594.
- Software Heritage ID: swh:1:dir:9c16bb0b0ba8693b09499d43ea90e4958cafac6a.

## DATA AVAILABILITY

Snapshots of the code are available in Software Heritage [28].

## LIST OF ABBREVIATIONS

AMI, Amazon Machine Image; AWS, Amazon Web Services; CLI, command line interface; HPC, high-performance computing; IaaS, Infrastructure as a Service; IaC, infrastructure as code; IP, Internet Protocol; IT, Information Technology; PaaS, Platform as a Service; SDK, Software Development Kits; WE, Workshop Environment.

## DECLARATIONS

### Ethical approval

Not applicable.

### Competing interests

The author(s) declare that they have no competing interests.

### Authors' contributions

JBC: Conceptualization, Investigation, Methodology, Software, Validation, Writing – original draft, Writing – review & editing. EG: Writing – review & editing. JPJC: Conceptualization, Funding acquisition. ER: Conceptualization, Funding acquisition, Project administration, Supervision, Resources, Writing – review & editing.

### Funding

This work was supported by the UK Research and Innovation (UKRI) scholars award, project reference MR/V038680/1, and the Natural Environment Research Council (NERC), project references NE/X006999/1 and NE/Y003527/1.

## Acknowledgements

The authors are pleased to acknowledge the invaluable feedback from Dr. Emma J. Barnes (University of York - Information Technology (IT) Services - Research IT), Richard Fuller (University of York - IT Services), Stuart Lacy (University of York - Research Software Engineering Team), Killian Murphy (University of York - IT Services), and Rosa Vicente (University of York - School of Physics, Engineering and Technology). Their participation in the first workshop on how to use the scripts and their feedback were key for the scripts, the online tutorial, and the workshops organisation to be much improved.

The authors are deeply grateful to the reviewers for their thorough revision of the paper and the detailed comments to improve the scripts and the documentation. We feel others will more easily benefit from using the scripts thanks to the reviewers' suggestions.

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
