## [Editor Report]

Editor’s AssessmentCloud computing services such as Amazon Web Services (AWS) provide opportunities to provide training in genomics and metagenomics analysis workflows without the need for participants to manage complex software installations or store large datasets in their computers. This work presents a set of Bash scripts to automate the management of AWS instances for use in such training workshops, making it easy and convenient to manage AWS instances for training delivery. Once an AWS account and a terminal environment have been configured, only two or three files need to be configured to create and manage instances for a workshop. The reviewers requested some more test data and examples in the paper, and upon testing demonstrated the scripts worked. As presented the scripts to support the delivery of ’omics training, but the scripts will work equally well for AWS Linux Ubuntu instances configured for other purposes.Editor’s AssessmentCloud computing services such as Amazon Web Services (AWS) provide opportunities to provide training in genomics and metagenomics analysis workflows without the need for participants to manage complex software installations or store large datasets in their computers. This work presents a set of Bash scripts to automate the management of AWS instances for use in such training workshops, making it easy and convenient to manage AWS instances for training delivery. Once an AWS account and a terminal environment have been configured, only two or three files need to be configured to create and manage instances for a workshop. The reviewers requested some more test data and examples in the paper, and upon testing demonstrated the scripts worked. As presented the scripts to support the delivery of ’omics training, but the scripts will work equally well for AWS Linux Ubuntu instances configured for other purposes.

---

## [Reviewer Report]

Reviewer name and names of any other individual's who aided in reviewerSindiswa LukheleDo you understand and agree to our policy of having open and named reviews, and having your review included with the published manuscript. (If no, please inform the editor that you cannot review this manuscript.)YesIs the language of sufficient quality?YesPlease add additional comments on language quality to clarify if neededIs there a clear statement of need explaining what problems the software is designed to solve and who the target audience is? YesAdditional CommentsThe statement of need is clear. Just a few questions: Does the script accommodate users new to AWS and without any form of training? It needs to be indicated in the paper. Is the source code available, and has an appropriate Open Source Initiative license <a href="https://opensource.org/licenses" target="_blank">(https://opensource.org/licenses)</a> been assigned to the code?YesAdditional CommentsAs Open Source Software are there guidelines on how to contribute, report issues or seek support on the code?YesAdditional CommentsIs the code executable?Unable to testAdditional CommentsIs installation/deployment sufficiently outlined in the paper and documentation, and does it proceed as outlined?Unable to testAdditional CommentsIs the documentation provided clear and user friendly?YesAdditional CommentsAdditional CommentsIs there a clearly-stated list of dependencies, and is the core functionality of the software documented to a satisfactory level?YesAdditional CommentsHave any claims of performance been sufficiently tested and compared to other commonly-used packages? Not applicableAdditional CommentsIs test data available, either included with the submission or openly available via cited third party sources (e.g. accession numbers, data DOIs)?NoAdditional CommentsThere was no biological data included in the paper. Probably the script needs to be tested using biological data. Are there (ideally real world) examples demonstrating use of the software? YesAdditional CommentsPlease add examples of the script using biological data.Additional CommentsAny Additional Overall Comments to the AuthorOverall, the paper is well written. A few things need to be considered, including using biological data as examples of how to run the script. Demonstrating using biological data will assist the user in following through with the examples, especially if there is no available training. RecommendationAccept

---

## [Reviewer Report]

Reviewer name and names of any other individual's who aided in reviewerGeert van GeestDo you understand and agree to our policy of having open and named reviews, and having your review included with the published manuscript. (If no, please inform the editor that you cannot review this manuscript.)YesIs the language of sufficient quality?YesPlease add additional comments on language quality to clarify if neededIs there a clear statement of need explaining what problems the software is designed to solve and who the target audience is? YesAdditional CommentsThe authors could stress the strengths of using cloud services for teaching a bit more, e.g.: low costs, self-managed, flexibility. Is the source code available, and has an appropriate Open Source Initiative license <a href="https://opensource.org/licenses" target="_blank">(https://opensource.org/licenses)</a> been assigned to the code?YesAdditional CommentsYes. However, in the manuscript text it is mentioned a CC-BY 4 license is used (which would not be very appropriate for software), while in the github repository there is an MIT license (https://github.com/Cloud-SPAN/aws-instances). I would suggest the authors to use the MIT license for the code and a CC-BY for the tutorial. As Open Source Software are there guidelines on how to contribute, report issues or seek support on the code?NoAdditional CommentsThe repository would benefit from instructions on how to contribute, e.g. in a CONTRIBUTING.md file. Is the code executable?Unable to testAdditional CommentsThe software requires a domain (as far as I understood). At time of review I wasn't in the capacity to register one. It would help if the authors would provide a quick 'getting started' that can all be performed with an AWS free tier. Is installation/deployment sufficiently outlined in the paper and documentation, and does it proceed as outlined?Unable to testAdditional CommentsSee above. However, the paper gives a broad overview, and there is a detailed tutorial on how to perform all the steps. Is the documentation provided clear and user friendly?NoAdditional CommentsThe tutorial is very detailed. However: - There is no link in the repository to the tutorial - The script works with configuration files as input. I found it hard to find out which options in e.g. resourcesIDs.txt were required.  - The documentation page (now README.md?) could use some structure and detail Is there enough clear information in the documentation to install, run and test this tool, including information on where to seek help if required?NoAdditional CommentsAlmost everything is there, however things are partly found in the tutorial. Is there a clearly-stated list of dependencies, and is the core functionality of the software documented to a satisfactory level?NoAdditional CommentsI think aws-cli is the only dependency, and that is probably stated in the tutorial, but there's e.g. no 'installation' header in README.mdHave any claims of performance been sufficiently tested and compared to other commonly-used packages? Not applicableAdditional CommentsIs test data available, either included with the submission or openly available via cited third party sources (e.g. accession numbers, data DOIs)?NoAdditional CommentsBut not really applicable. There is an example for inputs in the repository. Are there (ideally real world) examples demonstrating use of the software? YesAdditional CommentsIs automated testing used or are there manual steps described so that the functionality of the software can be verified?NoAdditional CommentsSome basic tests without having to interact with a personal account would be possible. Any Additional Overall Comments to the Author- As far as I could tell most (if not all) steps could be done with infrastructure as code (e.g. Terraform/Ansible). This is a general format that is used by many people. Can the authors state what the advantages of using only bash are over iac? - The configuration files in the input directory are plain text files. Consider to use one file with markup language like json or yaml. - A schematic overview of the resulting infrastructure including instances, network, keys/users/ and disks would be helpful for the reader - The 'Statement of need' hardly contains references to peer-reviewed literature. Although I don't think this should be a hard requirement, I do think it would make the manuscript stronger. Use e.g. existing literature on (bioinformatics) education, e.g. https://scholar.google.com/scholar?hl=nl&as_sdt=0%2C5&q=bioinformatics+teaching&btnG=&oq=bioinformatics+teaching - Make sure the user finds all documentation/tutorials. Cross reference between the repository and the tutorial.  - Suggestion: allow for mounting a shared disk. This enables learners to share files in e.g. group work.  - Suggestion: make as many options as possible optional, e.g. the domain, so all steps can be done with an AWS free tier. RecommendationMinor Revisions

---

## [Reviewer Report]

Reviewer name and names of any other individual's who aided in reviewerToby HodgesDo you understand and agree to our policy of having open and named reviews, and having your review included with the published manuscript. (If no, please inform the editor that you cannot review this manuscript.)YesIs the language of sufficient quality?YesPlease add additional comments on language quality to clarify if neededIs there a clear statement of need explaining what problems the software is designed to solve and who the target audience is? YesAdditional CommentsIs the source code available, and has an appropriate Open Source Initiative license <a href="https://opensource.org/licenses" target="_blank">(https://opensource.org/licenses)</a> been assigned to the code?YesAdditional CommentsThe repository containing the scripts includes an MIT license and a CITATION.cff file, which is very good practice. However, the manuscript (and Zenodo record) currently states that "The scripts are freely available to download and use under a Creative Commons BY 4.0 attribution license" -- this sentence and the Zenodo record should be corrected to reflect the MIT license of the software.As Open Source Software are there guidelines on how to contribute, report issues or seek support on the code?NoAdditional CommentsScripts include contact information for the corresponding author, and the CITATION.cff includes contact details for all authors. However, the repository contains no contributing guide, or guidance in the README.md, that could help would-be contributors understand how to get involved or contribute most effectively. The project README and the tutorial mentioned in the manuscript focus on usage of the scripts.Is the code executable?Unable to testAdditional CommentsThe scripts run, but I was unable to test them fully because of the rigidity of the required cloud environment configuration. Due to internal constraints the AWS environment I am working with could not be adjusted to fit exactly with the specifications of the authors' system in time for this review to be filed. Specifically, we could not configure a subdomain for the cloud instances created, and the way we handle security groups is also different. Although neither of these differences would prevent cloud instances from being created, the way the csinstances_create script cannot run without them. I note that the script is written with no default values set, and with the assumptions that 1. all of the required parameters will be included in the resourcesIDs.txt file and 2. that these parameters will appear in a fixed order. I believe it would be reasonable to allow users to run the scripts without having first created a hosted zone for the instances that will be created. For example, by adjusting the script to use default values where possible if parameters have not been set in the resourcesIDs.txt file. It would also be helpful to allow users to specify parameters in an arbitrary order within the resourcesIDs.txt file. Furthermore, I recommend that the authors explore the use of Launch Templates (https://docs.aws.amazon.com/AWSEC2/latest/UserGuide/ec2-launch-templates.html) to specify defaults for launched instances, which may prove simpler and more robust than the current approach of reading parameters from a text file and substituting those into a call to `ec2 run-instances`.Is installation/deployment sufficiently outlined in the paper and documentation, and does it proceed as outlined?YesAdditional CommentsThe paper and online tutorial provide clear and thorough guidance on how to use the scripts, including details that anticipated many of the questions that I had about using the software. I encourage the authors to link directly from the source repository containing the scripts to the tutorial, to make it easier for would-be users to find the information they need on how to use and adapt the scripts.Is the documentation provided clear and user friendly?YesAdditional CommentsThe accompanying tutorial is clear and well-structured. As mentioned above, I recommend that more links are created from the scripts source repository and that tutorial site, to help potential users find the relevant documentation to follow.Is there enough clear information in the documentation to install, run and test this tool, including information on where to seek help if required?YesAdditional CommentsIs there a clearly-stated list of dependencies, and is the core functionality of the software documented to a satisfactory level?YesAdditional Commentshttps://cloud-span.github.io/cloud-admin-guide-2-managing-aws-instances/setup.html contains information about the Bash version required and good instructions about how to install/update it on different operating systems. The AWS CLI tool, also required, is not discussed on that page but installation is the topic of a section in the main body of the tutorial.Have any claims of performance been sufficiently tested and compared to other commonly-used packages? Not applicableAdditional CommentsIs test data available, either included with the submission or openly available via cited third party sources (e.g. accession numbers, data DOIs)?YesAdditional CommentsThe source repository includes example config files that largely fulfil the purpose of example data. The only potential difficulty with these files is that they will only work with the authors' local AWS setup. I cannot think of a way that this could be avoided, however: execution of the scripts inevitably requires the accompanying AWS account setup and config. Are there (ideally real world) examples demonstrating use of the software? YesAdditional CommentsThe use of the software is fully described within the accompanying tutorial.Is automated testing used or are there manual steps described so that the functionality of the software can be verified?NoAdditional CommentsI suspect it would be difficult to create meaningful automated tests for these scripts, as they rely on interacting with the Amazon Web Services API to run. Any Additional Overall Comments to the AuthorI was delighted to receive this paper for review: the authors are describing automation of a process we have been handling manually for several years. The documentation accompanying the scripts is excellent: it is detailed, easy to follow, and comprehensive. I strongly recommend creating clearer links to that tutorial from the software repository on GitHub. Unfortunately, I was unable to test the complete workflow of the scripts as I could not access an AWS environment configured to the exact specifications described by the Cloud-SPAN team. If the authors are willing to adjust the scripts to be more permissive of alternative configurations (e.g. dropping the hard requirement for a subdomain where the instances could be hosted), I would be more than happy to review the new version. Thank you very much for writing the scripts, the documentation, and the paper -- and even more thanks for doing it all in the open, maximising the impact your work can have on the wider community.RecommendationMinor Revisions